# Investigating the Pathogenic Interplay of Alpha-Synuclein, Tau, and Amyloid Beta in Lewy Body Dementia: Insights from Viral-Mediated Overexpression in Transgenic Mouse Models

**DOI:** 10.3390/biomedicines11102863

**Published:** 2023-10-22

**Authors:** Melina J. Lim, Suelen L. Boschen, Aishe Kurti, Monica Castanedes Casey, Virginia R. Phillips, John D. Fryer, Dennis Dickson, Karen R. Jansen-West, Leonard Petrucelli, Marion Delenclos, Pamela J. McLean

**Affiliations:** 1Department of Neuroscience, Mayo Clinic, 4500 San Pablo Road, Jacksonville, FL 32224, USA; lim.melina@mayo.edu (M.J.L.); souza.suelen@mayo.edu (S.L.B.); kurti.aishe@mayo.edu (A.K.); castanedescasey.monica@mayo.edu (M.C.C.); phillips.virginia@mayo.edu (V.R.P.); dickson.dennis@mayo.edu (D.D.); jansenwest.karen@mayo.edu (K.R.J.-W.); petrucelli.leonard@mayo.edu (L.P.); delenclos.m@gmail.com (M.D.); 2Department of Neurosurgery, Mayo Clinic, 4500 San Pablo Road, Jacksonville, FL 32224, USA; 3Department of Neuroscience, Mayo Clinic, 13400 E. Shea Blvd, Scottsdale, AZ 85259, USA; fryer.john@mayo.edu

**Keywords:** Lewy body dementia, Lewy body, alpha-synuclein, tau, amyloid beta, mouse model, tail vein, AAV, Parkinson’s disease, Alzheimer’s disease

## Abstract

Lewy body dementia (LBD) is an often misdiagnosed and mistreated neurodegenerative disorder clinically characterized by the emergence of neuropsychiatric symptoms followed by motor impairment. LBD falls within an undefined range between Alzheimer’s disease (AD) and Parkinson’s disease (PD) due to the potential pathogenic synergistic effects of tau, beta-amyloid (Aβ), and alpha-synuclein (αsyn). A lack of reliable and relevant animal models hinders the elucidation of the molecular characteristics and phenotypic consequences of these interactions. Here, the goal was to evaluate whether the viral-mediated overexpression of αsyn in adult hTau and APP/PS1 mice or the overexpression of tau in Line 61 hThy1-αsyn mice resulted in pathology and behavior resembling LBD. The transgenes were injected intravenously via the tail vein using AAV-PHP.eB in 3-month-old hThy1-αsyn, hTau, or APP/PS1 mice that were then aged to 6-, 9-, and 12-months-old for subsequent phenotypic and histological characterization. Although we achieved the widespread expression of αsyn in hTau and tau in hThy1-αsyn mice, no αsyn pathology in hTau mice and only mild tau pathology in hThy1-αsyn mice was observed. Additionally, cognitive, motor, and limbic behavior phenotypes were not affected by overexpression of the transgenes. Furthermore, our APP/PS1 mice experienced premature deaths starting at 3 months post-injection (MPI), therefore precluding further analyses at later time points. An evaluation of the remaining 3-MPI indicated no αsyn pathology or cognitive and motor behavioral changes. Taken together, we conclude that the overexpression of αsyn in hTau and APP/PS1 mice and tau in hThy1-αsyn mice does not recapitulate the behavioral and neuropathological phenotypes observed in LBD.

## 1. Introduction

Lewy body dementia (LBD) describes a group of neurological disorders overlapping with Alzheimer’s disease (AD) and Parkinson’s disease (PD) in the clinical presentation of dementia and motor deficits, including PD dementia (PDD) and dementia with Lewy body (DLB) [1]. LBD is primarily classified as an alpha-synucleinopathy with the major neuropathological hallmark being the presence of alpha-synuclein (αsyn)-containing Lewy bodies (LBs) in the cortex, brainstem, and limbic areas. However, pathologically and biochemically, LBD appears to fall somewhere along a disease spectrum ranging from AD to PD since amyloid plaques containing amyloid-beta (Aβ) and tau lesions with hyper-phosphorylated tau often coexist in PDD and DLB brains [2,3,4,5]. For this reason, about 50% of all LBD patients having a sufficient AD-like pathology at autopsy receive a secondary neuropathologic diagnosis of AD [6]. The co-occurrence of αsyn, Aβ, and tau suggests a potential synergistic interaction of the individual pathologies that determines susceptibility to developing LBD.

The significance of AD and PD co-pathologies is not fully understood but is thought to contribute to the speed of progression and overall survival. Studies suggest that Aβ and αsyn have synergistic effects on symptoms in LBD due to a direct interaction between Aβ and αsyn [7,8]. Interestingly, a large multicenter longitudinal study of PD/DLB patients found that more than 70% of DLB patients had medium to high levels of coexisting AD neuropathologic changes at autopsy [9]. Additionally, roughly 50% of sporadic or familial AD cases have an αsyn pathology in the amygdala or other limbic regions at autopsy, with the exception of presenilin 1 and 2 (*PSEN 1* and *2*) mutation cases where LB pathology in the amygdala was found in 96% of the cases [10]. 

Similarly, the αsyn and tau synergistic interaction could contribute to the formation of disease-specific aggregates. It is known that αsyn is prone to self-aggregate, whereas tau cannot aggregate by itself and requires an inducing agent [11]. Considering that αsyn and tau can physically interact [12], it has been demonstrated that αsyn and tau promote each other’s aggregation in vitro, with tau accelerating αsyn oligomerization and αsyn acting as an inducing agent of tau aggregation through its hydrophobic non-amyloid component [13]. Moreover, tau expression enhances the toxicity and secretion of αsyn and changes the pattern of αsyn aggregation by promoting the formation of smaller inclusions in cellular models [14]. Additionally, phosphorylated tau aggregates have been reported in numerous synucleinopathy mouse models [15,16,17] reinforcing the idea of a synergistic interaction between αsyn and tau.

Further progress in LBD and in the interaction between αsyn, Aβ, and tau research is hindered by a lack of convincing and relevant animal models that could inform on the discovery and development of new therapeutics. The development of models recapitulating the comorbid pathology and differential clinical symptom onset of PDD and DLB will allow for a better understanding of LBD pathogenesis and the cellular mechanisms that lead to neurodegeneration. Thus, the goal of this study was to develop novel AAV-rodent models based on the pathogenic interplay of αsyn, Aβ, and tau, to identify the functional effect of the interaction of these proteins in the progression of the pathology observed in LBD. We hypothesized that introducing tau or αsyn via AAV in adult transgenic PD and AD lines would exacerbate the behavioral deficits and pathology associated with LBD. 

Here, we induced the widespread central nervous system (CNS) transgene expression of tau or αsyn via an AAV9-derived PHP.eB-viral-vector-mediated approach, delivered peripherally via an intravenous (IV) route [18,19]. Mice overexpressing mutant amyloid precursor protein (APP) and *PSEN 1* (APP/PS1) [20,21] or humanized tau (hTau) [22] were transduced with AAV-expressing human wild-type αsyn (αsyn), and mice expressing human αsyn under the Thy-1 promoter (line 61)(hThy1-αsyn) [23,24] were transduced with AAV expressing human 4R tau (tau). We predicted that the expression of transgenes, αsyn or tau, would create co-pathology that would translate into specific behavioral deficits.

We report that our models failed to mimic the symptoms observed in LBD patients, and only mild neuropathology could be observed. Here, we present data demonstrating that the IV introduction of αsyn and tau in three different transgenic animals where modest pathology is already present (αsyn, Aβ, and tau) does not reproduce behavioral and histopathological deficits that mimic symptoms and neuropathology in LBD patients.

## 2. Materials and Methods

### 2.1. Viral Vector Production

The Adeno-associated viral (AAV) vectors AAV-PHP.eB-venus, AAV-PHP.eB-tau, AAV-PHP.eB-syn were constructed from AAV serotypes 2/9 produced via plasmid transfection with helper plasmids in HEK293T cells. Then, cells were harvested and lysed in the presence of 0.5% sodium deoxycholate and 50 U/mL of Benzonase (Sigma-Aldrich, St. Louis, MO, USA) through freeze–thawing, and the AAVs were isolated using a discontinuous iodixanol gradient and buffer exchanged to PBS using an Amicon Ultra 100 Centrifugation device (Millipore, Burlington, MA, USA). Finally, the genomic titer of each virus was determined via quantitative polymerase chain reaction. 

### 2.2. Animals 

All procedures were in accordance with the Mayo Clinic Institutional Animal Care and Use Committee (IACUC) regulations. Mice were maintained on a 12 h light/dark cycle with food and water available ad libitum. *DBA-Tg (Thy1-SNCA)61Ema “Line 61”* (hThy1-αsyn) mice were received from the University of California, Los Angeles (Los Angeles, CA, USA). *B6.Cg-Mapt^tm1(EGFP)Klt^ Tg(MAPT)8cPdav/J* (hTau) and *B6.Cg-Tg(Thy1-APPSw, Thy1-PSEN1*L166P)21Jckr* (APP/PS1) mice were obtained from Jackson Laboratory (Bar Harbor, ME, USA). All lines were bred in house thereafter. 

hThy1-αsyn mice overexpress full-length human wild-type hαsyn under the Thy1 promotor and mimic the sporadic tendencies of PD, such as the progressive loss of striatal dopaminergic transmission. Although no nigral neuronal loss is achieved, αsyn pathological accumulation and motor and non-motor deficits are measurable as early as 4 months of age, making this mouse a useful PD model [23,25]. hTau mice lack endogenous mouse microtubule-associated protein tau (*MAPT*), but instead express the full-length human *MAPT* gene under the endogenous human *MAPT* promoter [22]. hTau mice develop the AD-related spatiotemporal distribution of hyper-phosphorylated tau at around 3 months of age, which leads to the formation of insoluble tau aggregates beginning at approximately 6 months of age and exhibit modest behavioral abnormalities with age. Later in life (>15 months), thioflavine-S-positive neurofibrillary tangles (NFTs) have also been reported [22,26]. This transgenic line is an attractive model for studying tauopathy as these mice develop tangles without introducing tau mutations. The double transgenic APP/PS1 mice express both the chimeric mouse and Swedish human APP mutation along with mutant human *PSEN 1*. Not only are both mutations linked to early-onset AD, these mice are also known to develop Aβ plaques as early as 6 months of age [27].

### 2.3. Experimental Design

Each transgenic line had three treatment groups consisting of one non-transgenic (NTG) treated with AAV-PHP.eB-wtsyn (NTG + syn) or AAV-PHP.eB-tau (NTG + tau) and two transgenic (TG) groups, in which one received AAV-PHP.eB-wtsyn (TG + syn) or AAV-PHP.eB-tau (TG + tau) and the other received AAV-PHP.eB-venus (TG + venus) (Figure 1A). The viral construct containing the yellow fluorescent protein venus was used as an experimental control for the viral-mediated transgene (αsyn, tau) expression. 

All mice were tail-vein injected at 3 months of age with 16 × 10^10^ genome copies/milliliter (gc/mL) of the virus and transgenes (AAV-PHP.eB-wtsyn and AAV-PHP.eB-tau). Animals underwent a battery of behavioral assessments at 3 months post-injection (3-MPI), 6-MPI, or 9-MPI. After the final day of behavioral tests, mice were euthanized, and their brains were removed for histological analyses (Figure 1B). 

### 2.4. Behavioral Assessments

Behavioral assessments were performed by experimenters blinded to the genotype and transduction status of the mice. Mice were brought to the testing room 1–2 h prior to the beginning of the behavioral tests to allow for proper acclimation, and all equipment was cleaned with 30% ethanol in between each animal. Each group of mice (n = ±30) underwent a series of behavioral tests to assess their exploratory and anxiety-like behavior (open field, elevated plus maze), cognitive function (fear conditioning), and motor ability (pole test and beam walk) in the respective order.

#### 2.4.1. Open Field Test 

Mice were placed in the center of an open-field arena (40 × 40 × 30 cm, W × L × H) for 15 min with mounted cameras above to record activity. Immediately following the test, mice were returned to their home cage and arenas were cleaned. To evaluate motor functions, total distance travelled, speed, and time spent in the center were measured. Quantification was performed utilizing ANYmaze software v7.2 (Stoelting Co., Wood Dale, IL, USA).

#### 2.4.2. Elevated Plus Maze

The maze is elevated roughly 50 cm above the floor and consists of two open arms (50 × 10 cm) and two high opaque closed arms (50 × 10 × 40 cm). Mice were placed in the center of the maze, front facing an open arm, and allowed to explore the apparatus for 5 min. Anxiolytic behavior was measured with time spent in open and closed arms and the ratio between open vs. closed. Movements were tracked with ANYmaze software.

#### 2.4.3. Fear Conditioning

Mice were allowed to explore a sound-attenuated grid floor chamber for 1 min, followed by a 30 s white noise (80 dB) (conditioned stimulus (CS)) that was accompanied by a mild foot shock (0.2 mA) in the last 2 s (unconditioned stimulus (US)). This CS–US pair was administered again after 1 min; then, 30 s later mice were returned to their home cage overnight. The following day (18–24 h later), mice were placed back into the same chamber, and freezing behavior was recorded for 5 min in the absence of CS and US presentation to test contextual learning. To test cued learning, environmental and contextual cues were altered by placing a partition diagonally in the chamber with vanilla extract on one side, changing white to red lights, and wire to plastic flooring. The CS was delivered without a foot shock for 3 min, and mouse freezing behavior was recorded. 

#### 2.4.4. Pole Test 

A wooden dowel (0.7 × 50.8 cm, D × H) was attached to a base (12.7 × 11.8 × 1.8 cm, W × L × H) that was roughly sanded once before training. The pole was wiped down with 30% ethanol before and after each cage and was placed in the center of the cage bottom. For training, mice were placed individually at the top of the pole upright with the head towards the ceiling and were gently guided to make a T-turn and climb down the pole five times each. This was performed for two consecutive days followed by the test session later in the afternoon on the second day. For the test session, mice were placed on top of the pole, but no assistance was provided. The mouse’s ability to perform a T-turn and climb down five times was recorded. 

#### 2.4.5. Beam Walk 

A wooden dowel (0.3 × 80 cm, D × L) was roughly sanded once before training and secured approximately 50 cm above the floor with the assistance of a support stand. The mouse’s home cage was placed at one end of the beam as an incentive to cross. Mice received one day of training with assistance when necessary to cross the beam. Each mouse was tested five times, and the time each animal took to cross the beam, as well as the number of slips, was recorded. 

### 2.5. Exsanguination and Tissue Preparation 

All animals were deeply anesthetized with a ketamine/xylazine cocktail (100/20 mg/kg) prior to undergoing a transcardial perfusion and exsanguination with 1× Phosphate Buffer Saline (PBS). The brains were removed and bisected along the midline. One half was drop-fixed overnight at 4 °C in 4% paraformaldehyde and then subsequently embedded in paraffin wax and mounted on glass slides at 5 µm in sagittal sections. 

### 2.6. Immunohistochemistry 

The tissue was rehydrated in a graded series of xylene and alcohols and steamed with distilled water for antigen retrieval. Sections were stained for αsyn (4B12, Biolegend, San Diego, CA, USA; 5G4, Millipore, Burlington, MA, USA), phosphorylated-αsyn pS129 (pSyn#64, WAKO, Richmond, VA, USA), tau (E1, [kind gift from Dr. Len Petrucelli and Dr. Casey Cook]), phosphorylated-tau pS202, Thr205 (AT8, Invitrogen, Waltham, MA, USA), and amyloid plaque (Thioflavine S, Sigma-Aldrich, St. Louis, MO, USA) and were visualized using the Envision Plus system (DAKO, Carpinteria, CA, USA). Counterstaining was performed with hematoxylin and 1× Scott’s tap water. The slides were subsequently dehydrated in a graded series of alcohols and xylenes and were coverslipped. All slides were scanned with the Aperio AT2 brightfield scanner (Deer Park, IL, USA) with a magnification of 20×. Immunofluorescence was scanned utilizing the Keyence BZ-X800E (Itasca, IL, USA) with a magnification of 20×. These original scans were used for final quantification, and then, images were zoomed in to allow for a close-up visualization of protein labeling. 

### 2.7. Statistics 

GraphPad Prism v9.2 (San Diego, CA, USA) was utilized for data analysis. Immunohistochemistry and behavioral data from elevated plus maze, beam walk, open field, and pole tests assumed a non-parametric distribution and were analyzed using the Kruskal–Wallis and Dunn’s test for multiple comparisons. Data from the contextual and cued fear conditioning assumed a parametric distribution and were analyzed with one-way ANOVA followed by Tukey’s test for multiple comparisons. To evaluate whether sex influenced the expression of tau and αsyn in the different transgenic and non-transgenic lines, we overruled the non-normal distribution of the data and conducted two-way ANOVA and Tukey’s post-hoc test. Differences were considered statistically significant when *p* < 0.05. Results are presented as the median ± interquartile range or mean ± SEM according to the statistical method used.

## 3. Results

### 3.1. Systemic Administration of AAV-PHP.eB-Venus Induces Brain-Wide Venus GFP Expression 

Although the AAV-PHP.eB capsid can enter the CNS via LY6A receptors present in rodents, some mouse strains are not permissible to PHP.eB [28]. In this study, all mouse strains have a C57BL/6 × DBA background. To confirm that this mouse strain is permissible to the AAV-PHP.eB capsid, we injected both transgenic and non-transgenic littermates of hThy1-αsyn mice at 3 months of age with 200 μL of the virus (1.10 × 10^12^ gc/mL). After two weeks, mice were exsanguinated for brain collection, and the extent of transduction was visualized using fluorescent microscopy. The systemic administration of AAV-PHP.eB-venus induced widespread transduction of venus in the brain of hThy1-αsyn mice (Figure 1C), demonstrating that AAV-PHP.eB is an effective vector for carrying transgenes across the blood–brain barrier in C57BL/6 × DBA mice.

### 3.2. Overexpression of Human αsyn in hTau Mice Does Not Induce Behavior and Pathological DLB-like Phenotypes 

To evaluate the behavioral phenotypes of the hTau mice with the brain overexpression of αsyn, a battery of exploratory, motor, and cognitive tests was performed on mice at 6- and 9-MPI (see Appendix A for statistical details). Transgenic hTau mice overexpressing αsyn (TG + syn) did not develop significant motor deficits as assessed in the open field (Figure 2A) and beam walk (Appendix A) tests. Only a mild hypolocomotion in the open field was observed in hTau males expressing venus (TG + venus) at 6-MPI. The pole test evaluates basal ganglia-related locomotion by assessing the ability of mice to grasp and maneuver on a pole in order to descend to their home cage [29]. Interestingly, non-transgenic mice overexpressing αsyn (NTG + syn) took longer to descend the pole than TG + syn and TG + venus. Particularly, the NTG + syn females exhibited the highest time to descend the pole (Figure 2B). 

NTG + syn males showed decreased freezing behavior in the contextual fear conditioning compared to TG + syn and TG + venus, although no significance was reached (Figure 2C). Although changes in anxiety-like behavior were not observed in the EPM test (Appendix A), TG + syn mice spent more time in the center of the open field arena (Figure 2A), compared to TG + venus, suggesting lower anxiety levels. 

Despite the lack of strong motor, anxiety-like, and cognitive phenotypes, widespread expression of αsyn was observed in the thalamus (H = 14.65, *p* < 0.001), cortex (H = 14.93, *p* < 0.001), and hippocampus (H = 13.61, *p* < 0.01) of NTG + syn and TG + syn compared to TG + venus (Figure 3A). Alpha-syn pathology was not observed in the levels of phosphorylated αsyn (p-syn) (Figure 3B, thalamus: H = 4.16, *p* = 0.12; cortex: H = 0.63, *p* = 0.73; hippocampus: H = 5.78, *p* = 0.06). Additionally, TG + syn and TG + venus mice demonstrated increased tau expression compared to NTG + syn, but no significant difference between those two groups was observed (Figure 3C, thalamus: H = 35.24, *p* < 0.0001; cortex: H = 14.76, *p* < 0.001; hippocampus: H = 14.03, *p* < 0.001). Interestingly, male TG + syn mice, but not female TG + syn mice, expressed lower levels of tau in the thalamus and cortex, compared to the same sex of TG + venus (Figure 3C). This suggests that αsyn may regulate tau expression in male hTau mice, but not in hTau females. Higher levels of phosphorylated tau (p-tau) in the cortex of TG + syn mice, but not in TG + venus mice, were also observed (Figure 3D, thalamus: H = 3.09, *p* = 0.21; cortex: H = 5.94, *p* = 0.05; hippocampus: H = 1.69, *p* = 0.43).

### 3.3. Overexpression of Human Tau in hThy1-αsyn Mice Does Not Induce Behavior and Pathological DLB-like Phenotypes 

Because the overexpression of humanized αsyn in mice expressing all six isoforms of tau produced only minor changes in neuropathology, motor, and cognitive behavior at both 6- and 9-MPI and poorly recapitulated deficits observed in DLB, we decided to evaluate whether AAV-induced humanized tau overexpression in an established transgenic model of αsyn overexpression would aggravate or accelerate the behavior and pathology. In order to fully characterize potential dysfunctions over time, we added behavioral and histological evaluations at an earlier time-point (3-MPI). We show the complete results from 3-MPI in Figure 4 and Figure 5 and Appendix A. Statistical details from 3-, 6-, and 9-MPI are described in Appendix A. hThy1-αsyn transgenic mice overexpressing tau (TG + tau) did not have significant motor impairment as assessed in the open field (Figure 4A) and beam walk (Appendix A) tests, compared to NTG mice overexpressing tau (NTG + tau). In the pole test, female hThy1-αsyn mice injected with AAV + venus (TG + venus) took longer to descend the pole and to perform a T-turn than female NTG + tau and TG + tau mice, although high variability in the responses was observed between male and female TG + tau mice (Figure 4B). Additionally, both TG + tau and TG + venus females showed an increased freezing response in the contextual and cued fear conditioning tests (Figure 4C). No significant differences were observed regarding changes in the anxiety-like phenotype in the open field (Figure 4A), although in the EPM, NTG + tau demonstrated more time spent in the closed arms compared to TG + tau and TG + venus (Appendix A). In summary, our behavioral data suggest that hThy1-αsyn males may be more vulnerable to motor dysfunction, while females may be more vulnerable to cognitive dysfunction induced by αsyn expression, regardless of the co-expression of tau. 

The systemic injection of AAV-PHP.eB-tau induced the widespread brain expression of tau in NTG + tau and TG + tau mice relative to TG + venus (Figure 5A, thalamus: H (3, 29) = 13.37, *p* < 0.01, cortex: H (3, 29) = 11.26, *p* < 0.01, hippocampus: H (3, 29) = 11.29, *p* < 0.01). Levels of p-tau were increased in the hippocampus of NTG + tau mice only (Figure 5B, hippocampus: H (3, 28) = 9.10, *p* < 0.05; cortex: H (3, 28) = 2.62, *p* = 0.27; H (3, 28) = 3.63, *p* = 0.16). As expected, TG + tau and TG + venus expressed high levels of αsyn, but no significant differences in αsyn expression were observed between TG + tau and TG + venus (Figure 5C, thalamus: H (3, 35) = 20.15, *p* < 0.0001; cortex: H (3, 35) = 18.57, *p* < 0.0001; hippocampus: H (3, 35) = 18.06, *p* < 0.0001). Similarly, levels of p-syn were increased in the cortex, thalamus, and hippocampus of TG + tau and TG + venus mice (Figure 5D, thalamus: H (3, 34) = 12.71, *p* < 0.01; cortex: H (3, 34) = 12.30, *p* < 0.01; hippocampus: H (3, 34) = 12.11, *p* < 0.01). Although scattered increases in tau and αsyn expression in either male or female mice were observed, no significant differences were observed possibly due to the high variability of tau expression within each group (Figure 5).

### 3.4. Overexpression of Human αsyn in APP/PS1 Mice Does Not Induce Behavior and Pathological DLB-like Phenotypes

Considering that we did not recapitulate a DLB-like phenotype by inducing the overexpression of either tau or αsyn via the AAV-PHP.eB systemic delivery to mice transgenically expressing human αsyn or tau, we evaluated whether αsyn and Aβ could interact and aggravate the pathology. We used APP/PS1 mice tail-vein-injected with AAV-PHP.eB-wtsyn or AAV-PHP.eB-venus at 3-MPI. Our initial goal was to perform the battery of behavioral tests at 3-, 6-, and 9-MPI, as conducted with the other mouse strains. However, the APP/PS1 mice exhibited increased mortality after viral vector injection, precluding complete behavioral and histological analyses at all time points. Thus, we present only the data from animals that successfully concluded all analyses at 3-MPI (see Appendix A for statistical details). We did not observe significant differences in any of the exploratory, motor, and cognitive tests (Figure 6 and Appendix A), although the power of the statistical analyses was limited by the low number of animals per group (i.e., female TG + venus n = 2, female TG + syn n = 2, male TG + venus n = 2).

An immunohistochemical analysis of αsyn expression (Figure 7A) revealed little expression of this protein and its phosphorylated form (Figure 7B), with no significant differences among groups. Additional staining for amyloid plaques (ThioS) (Figure 7C) demonstrated no significant differences as well.

## 4. Discussion

Progress in LBD research would be significantly enhanced by the development of reliable animal models that efficiently mimic the pathological protein–protein interactions observed in this disease. The co-existence of αsyn and hyperphosphorylated tau is commonly observed in the brains of patients neuropathologically diagnosed with PDD and DLB [30], suggesting that these proteins may interact and aggravate symptomatology and neurodegeneration [31]. Here, we aimed to determine whether inducing the overexpression of human αsyn or tau in well-characterized, transgenic AD and PD mice models would exacerbate the existing phenotype and recapitulate the neurobiology of LBD. We successfully demonstrate the widespread brain expression of the transgenes αsyn and tau via the intravenous injection of AAV-PHP.eB in each of our transgenic mouse models, consistent with previous studies [18,19]. However, neither the hTau nor the hThy1-αsyn mouse models developed any earlier, additional, or exacerbated behavior or pathology with the co-expression of αsyn or tau, respectively. 

Previous studies on multiple cell lines have demonstrated that the co-expression of tau and αsyn produces a pathological synergistic effect in which one may influence the toxicity and aggregation of the other [14,32]. Additionally, it has been shown that the concentration of αsyn can contribute to increased tau-associated droplets in a liquid–liquid phase separation setting [33]. Despite increasing in vitro evidence of the pathological interaction of tau and αsyn, in vivo effects have been variable. Recently, it was suggested that αsyn may influence the rate of the tau burden, although the presence or absence of tau is not required for the development, spreading, or reduction of the αsyn pathology in mice [34]. Conversely, Vermilyea and colleagues (2022) demonstrated that in a hTg-A53T/mTau^−/−^ mouse model injected with human preformed fibrils (PFFs), the loss of tau delayed the onset of motor deficits, reduced αsyn pathology, and prolonged the lifespan of hTg-A53T mice, despite the lack of an effect of tau expression on αsyn aggregation [35]. In another study, however, tau reduction failed to prevent motor deficits and striatal dopaminergic terminal loss in a heterozygous transgenic mouse model expressing wild-type human αsyn [36]. A key factor contributing to the poor reproducibility of in vivo αsyn-tau pathological synergy is that different species of αsyn and tau may cause different levels and mechanisms of toxicity. Interestingly, studies that have used wild-type human αsyn and tau, including ours, failed to demonstrate significant changes in the phosphorylated forms, aggregates, and spreading of the proteins [37,38]. On the other hand, studies that used the point-mutations αsyn or tau and αsyn PFFs show some type of synergistic effect with at least one of the proteins [39,40]. Therefore, it is likely that a successful animal model of LBD is dependent on the seeding capacity of αsyn or tau to induce the formation of intracellular protein aggregates and behavioral deficits. 

Importantly, previous studies have suggested a potential interaction between Aβ and αsyn pathology [41,42,43,44]. It was shown that αsyn and Aβ may interact bidirectionally to accelerate the aggregation and spreading of αsyn pathology and hyperphosphorylated tau after the intra-hippocampal injection of αsyn PFFs in 5×FAD mice [34]. This suggests that Aβ oligomers potentiate the aggregation and spread of abnormal αsyn species through the brain, and both Aβ and αsyn pathology are necessary to drive the cross-seeding of tau [45]. Indeed, we attempted to use the same approach described in this study to investigate potential interactions between αsyn and Aβ. APP/PS1 mice were injected with AAV-PHP.eB-wtsyn, and behavioral and histochemical assessments at 3-MPI were performed. Unfortunately, we could not complete assessments at 6- and 9-MPI due to increased mortality starting at 3-MPI. However, the remaining animals at 3-MPI that survived were evaluated, with the results indicating no additional pathology with the overexpression of αsyn and pre-existing Aβ accumulation. Although minimal liver and renal toxicity has been demonstrated following AAV-PHP.eB systemic administration in C57BL/6J mice [46], it has been reported that the efficiency of viral transduction may be affected by the mouse strain [47,48]. Thus, different mouse strains may also have different susceptibilities to the hepatotoxic or nephrotoxic effects induced by AAV-PHP.eB. Moreover, Minkeviciene and colleagues (2009) reported sporadic seizures in the transgenic APP/PS1 mice but not wild-type littermates as early as 3 months, suggesting that the frequent and prolonged seizures contribute to the sudden deaths observed [49]. Further investigations are necessary to confirm whether APP/PS1 mice are more vulnerable to the systemic toxic effects induced by different serotypes of AAV and if the AAV may exacerbate the seizures that may occur. Additionally, further studies are needed to determine whether results would change with increased statistical power.

Limitations within our study have emphasized the importance of the αsyn and tau species that are being used to induce aggregation and pathology, as we only used wild-type human forms of these proteins. Additionally, the breadth of the study was hindered as our APP/PS1 mice experienced potential liver and kidney toxicity from the AAV-PHP.eB that may have exacerbated the predisposed nature of these mice. Finally, although widespread brain expression of the transgenes was achieved via intra-tail vein injection of the AAV-PHP.eB, other routes of administration, such as the retro-orbital sinus, may be more efficient to deliver a higher concentration of the virus to the CNS [18,47].

In summary, we demonstrate here that pre-existing αsyn and tau pathologies are not aggravated by the overexpression of wild-type human tau or αsyn in mice, possibly due to the absence of a seeding or inducing factor, such as αsyn PFFs. Additionally, we did not observe behavioral and histopathological deficits with the overexpression of wild-type human αsyn based on pre-existing Aβ plaques. Future studies will be necessary to explore the impact of different species of αsyn, tau, and Aβ in a potential synergistic pathology.

## Figures and Tables

**Figure 1 biomedicines-11-02863-f001:**
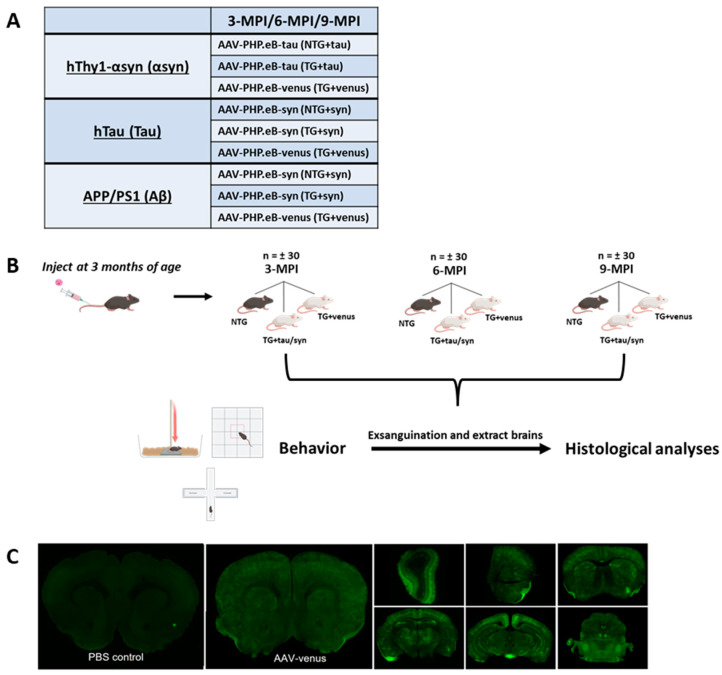
Experimental Paradigm (**A**) Table representing each transgenic line (hThy1-αsyn, hTau, and APP/PS1) and their respective treatment groups. NTG mice received AAV-PHP.eB-syn/tau (NTG + syn) or (NTG + tau), and TG mice received AAV-PHP.eB-syn/tau (TG + syn) or (TG + tau) and venus (TG + venus). (**B**) Mice were injected at 3-months-old and underwent behavioral testing and were subsequently euthanized 3-, 6-, 9- months post-injection (MPI). (**C**) Representative image of AAV-PHP.eB-venus transduction efficiency and ability to cross blood brain barrier in 3-month-old hThy1-αsyn mice 2 weeks post-injection versus PBS-injected hThy1-αsyn control mice.

**Figure 2 biomedicines-11-02863-f002:**
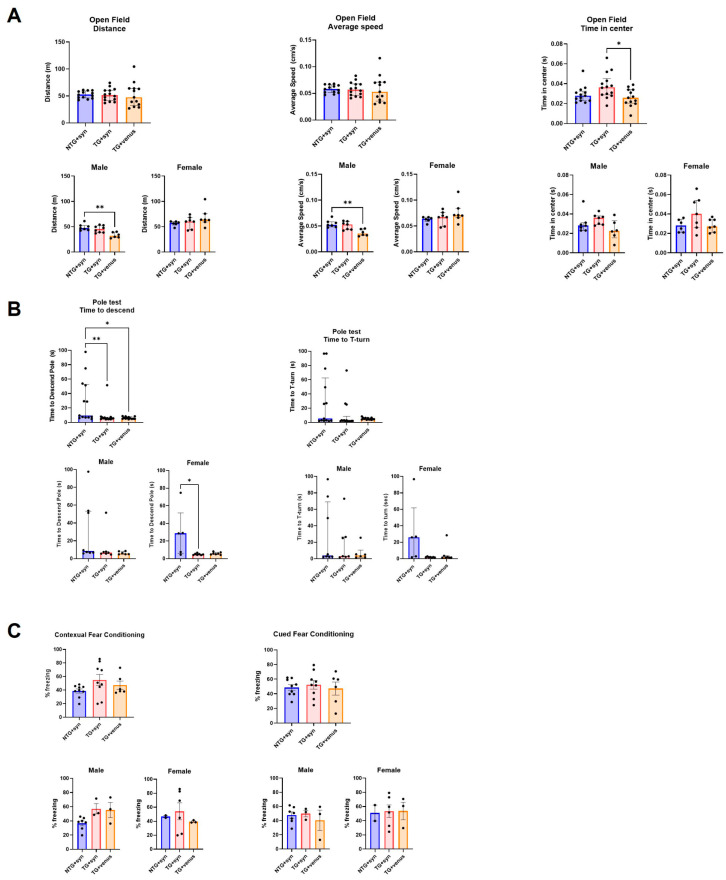
Motor behavior and spatial memory in 6-MPI hTau overexpressing human αsyn (**A**) An open field test demonstrated no significant differences observed in parameters measuring distance covered (left) and average speed (middle). However, transgenic hTau transduced with αsyn demonstrated significance in the time in center compared to TG + venus (right) [n = ±30, Kruskal–Wallis, Dunn’s test, * *p* < 0.05, ** *p* < 0.01]. (**B**) NTG + syn demonstrated a significant difference in time to descend the pole compared to TG + syn and TG + venus (left) but no significance in the T-turn (right) [n = ±30, Kruskal–Wallis, Dunn’s test, * *p* < 0.05]. (**C**) Both contextual fear memory (left) and cued fear memory (right) demonstrated no significance [one-way ANOVA, Tukey post-hoc, * *p* < 0.05].

**Figure 3 biomedicines-11-02863-f003:**
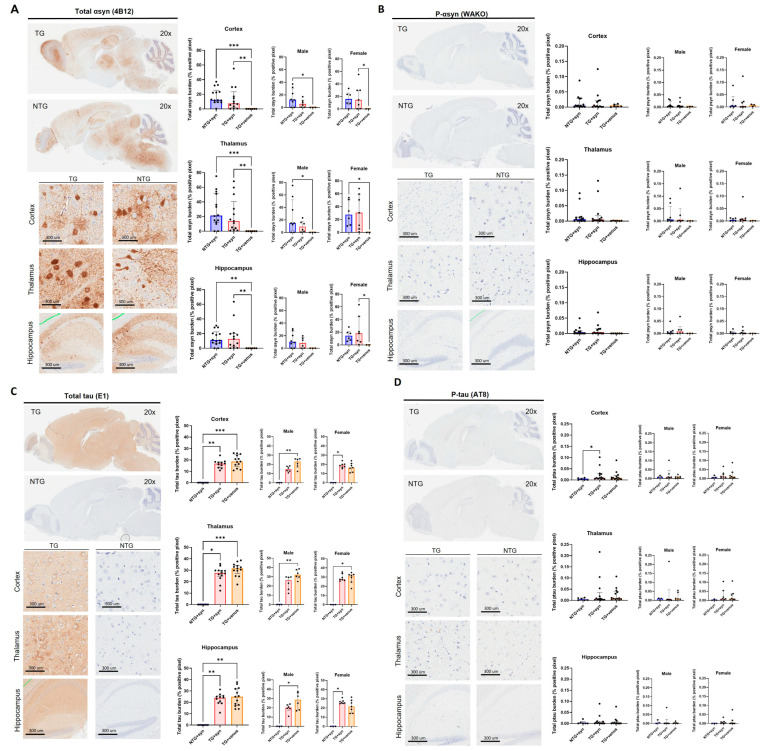
Transgene expression of human αsyn and human tau in 6-MPI hTau. Total αsyn staining indicated strong expression of the transgene that was also widely distributed in the brain [n = ±30, Kruskal–Wallis, Dunn’s test, * *p* < 0.05, ** *p* < 0.01, *** *p* < 0.001] (**A**). Despite the presence of αsyn, p-syn did not correlate these results [n = ±30, Kruskal–Wallis, *p* < 0.05] (**B**). Further staining of the total brain [n = ±30, Kruskal–Wallis, Dunn’s test, * *p* < 0.05, ** *p* < 0.01, *** *p* < 0.001] (**C**) and p-tau demonstrates no additional pathology with the presence of αsyn [n = ±30, Kruskal–Wallis, Dunn’s test, * *p* < 0.05] (**D**).

**Figure 4 biomedicines-11-02863-f004:**
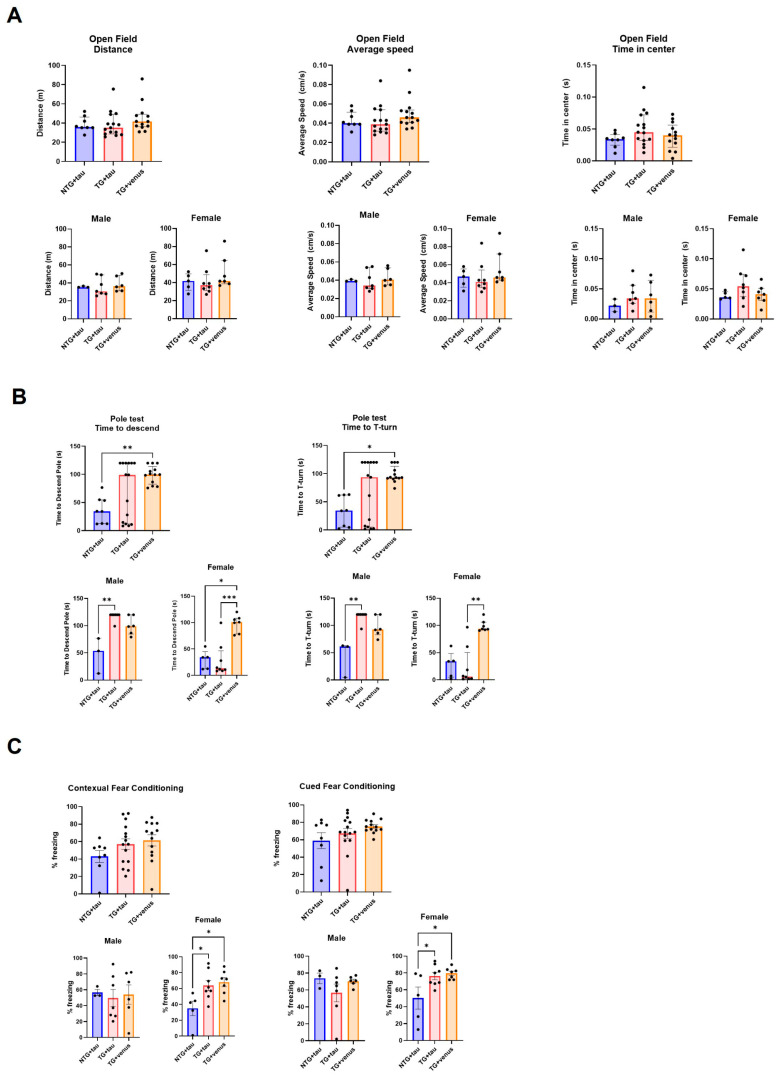
Assessment of motor behavior and procedural and spatial memory in 3-MPI hThy1-αsyn mice overexpressing human tau (**A**) Open field tests indicated no significant differences in the distance mice traveled (left), their average speed (middle), and time spent in the center (right) [n = ±30, Kruskal–Wallis, *p* < 0.05]. (**B**) Similar results were seen in the pole test, with no significant differences in the average time to descend the pole (left) and average time to the T-turn (right) [n = ±30, Kruskal–Wallis, * *p* < 0.05, ** *p* < 0.01, *** *p* < 0.001]. (**C**) Contextual (left) and cued (right) fear conditioning recapitulated these trends with no significance in all groups and treatments [n = ±30, one-way ANOVA, * *p* < 0.05].

**Figure 5 biomedicines-11-02863-f005:**
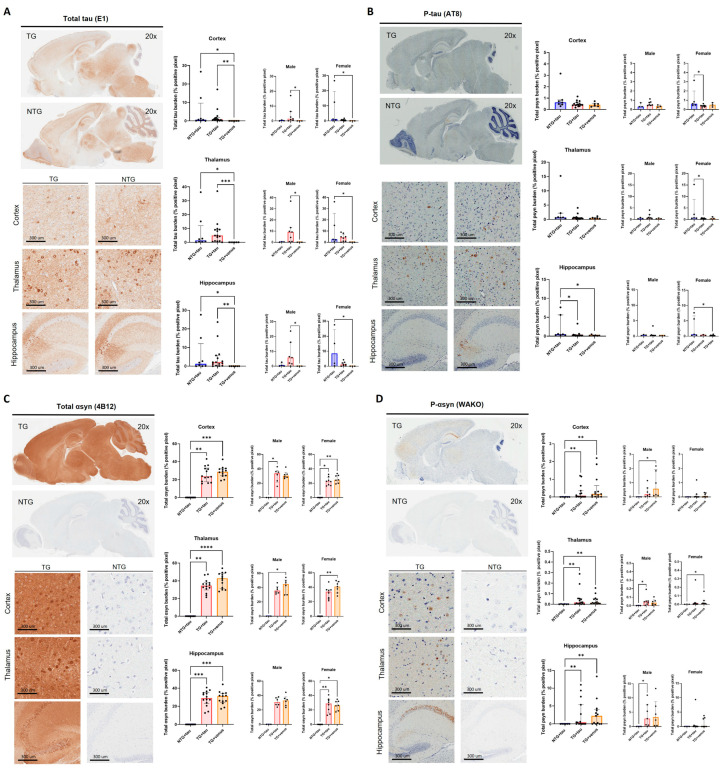
Brain-wide expression of total human tau and phosphorylated-tau in 3-MPI hThy1-αsyn mice. Immunohistochemistry for total tau [n = ±30, Kruskal–Wallis, Dunn’s test, * *p* < 0.05, ** *p* < 0.01, *** *p* < 0.001] (**A**) and p-tau (**B**) in a sagittal section [n = ±30, Kruskal–Wallis, Dunn’s test, * *p* < 0.05]. Images from the cortex, thalamus, and hippocampus demonstrate the transgene’s widespread expression throughout the brain. Semi-quantitative analyses of total αsyn [n = ±30, Kruskal–Wallis, Dunn’s test, * *p* < 0.05, ** *p* < 0.01, *** *p* < 0.001, **** *p* < 0.0001] (**C**) and p-syn (**D**) demonstrated no differences with the presence of overexpressed tau [n = ±30, Kruskal–Wallis, Dunn’s test, * *p* < 0.05, ** *p* < 0.01].

**Figure 6 biomedicines-11-02863-f006:**
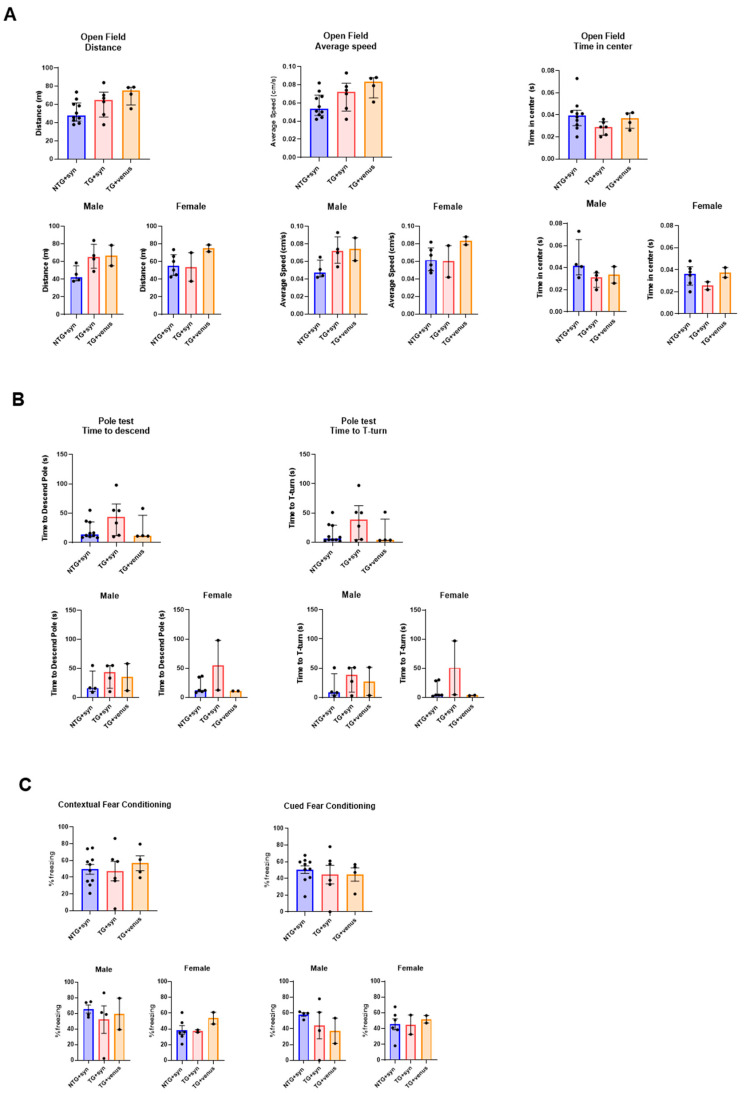
Motor and memory evaluation in 3-MPI APP/PS1 mice transduced with human αsyn. (**A**) No significant differences were observed in the open field test in the distance (left), speed (middle), and time spent in the center (right) [n = ±20, Kruskal–Wallis, Dunn’s test]. (**B**) The pole test indicated no significant differences in the time to descend (left) and T-turn (right) [n = ±20, Kruskal–Wallis, Dunn’s test] (**C**) This trend was also seen in the contextual (left) and cued (right) fear conditioning, with no significance [n = ±20 per MPI group, one-way ANOVA, Tukey post-hoc].

**Figure 7 biomedicines-11-02863-f007:**
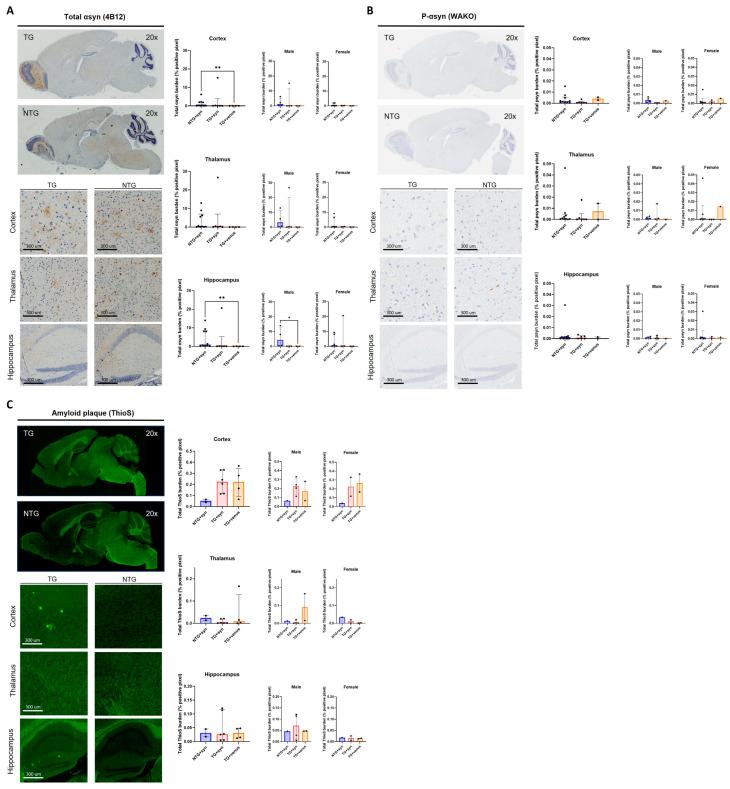
Immunohistochemistry of 3-MPI APP/PS1 transduced with human αsyn (**A**) Staining for αsyn revealed variable total and phosphorylated (**B**) synuclein, with no significant differences amongst groups in either stain [n = ±20, Kruskal–Wallis, Dunn’s test, * *p* < 0.05, ** *p* < 0.01]. Additional staining for amyloid plaques (**C**) also showed no significant differences [n = ±20, Kruskal–Wallis, Dunn’s test].

## Data Availability

The data presented in this study are available on request from the corresponding author. The data are not publicly available due to privacy concerns.

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
