# Peer review of "Investigating the Pathogenic Interplay of Alpha-Synuclein, Tau, and Amyloid Beta in Lewy Body Dementia: Insights from Viral-Mediated Overexpression in Transgenic Mouse Models"

_biomedicines, 2023, doi:10.3390/biomedicines11102863_

Round 1
Reviewer 1 Report
In this manuscript, the authors assess the effects of viral expression of alpha-synuclein on pathology in hTau and APP/PS1 AD mouse models. Mice were injected with the virus at 3 months of age and phenotypic and pathologic assessments were made at 6-, 9-, and 12-month time points. In addition, transgenic mice expressing alpha-synuclein under the Thy1 promoter were injected with a Tau-expressing virus. No behavioral effects and mild Tau pathology were found in the hTau mice overexpressing mice and APP/PS1 mice died 3 months post-injection. Overall, the authors do not find significant synergistic effects between alpha-synuclein and Tau or between alpha-synuclein and APP/Abeta using the described approaches to coexpress the various proteins in these experiments. The lethality of the AAV injection in APP/PS1 mice was well discussed. The experiments are well designed and rigorously analyzed and they address important current questions in comorbid neurodegenerative disease and should be published to inform subsequent efforts by the community to address these questions.
Author Response
We thank reviewer 1 for the time dedicated to review our manuscript and for the thoughtful comments.
Reviewer 2 Report
The article analyzes the effects of overexpression of αsyn in hTau and APP/PS1 mice and tau expression in hThy1-αsyn mice. Authors do not observe significant changes in behavioral and neuropathological phenotypes observed in LBD. The manuscript is interesting but I have some concerns:
1.- A section of limitations of the study should be added
2.- The objective used in the different IHQ and immunofluorescence should be indicated
Author Response
We thank Reviewer 2 for the time dedicated to review our study and we appreciate your comments to enhance the quality of our manuscript.
We have provided an additional paragraph in our Discussion to address the limitations of our study. We have also added to Figure 2, 4, and 6 the objectives used to image our immunohistochemical and immunofluorescence experiments.
Reviewer 3 Report
This was a well-executed study, despite generating results that were somewhat unexpected. The authors should be commended for their manuscript and willingness to publish their data.
I have provided some comments/suggestions that may improve manuscript.
Comments
In experimental design, could it be made clearer what each AVV is expressing including AAV-PHP.eB-venus?
This sentence is confusing - “All mice were tail-vein injected at 3-months of age with 200 μl of the virus and transgene with the same concentrations of viral load (8x1011gc/ml).”
Virus is not a volume (better to say how many viral particles inject in a 200µL volume throughout manuscript) and what’s the difference between virus, transgene and viral load???
Also, what does gc/ml stand for???
Fig 1 C legend: please provide more detail. Age of mice when injected and when post injection were images collected.
Results
Why not include 9 MPI data in Figure 2??
Could the behavioral test be named at the top of each graph?
Lines 288 and 292: should it be “TG+tau”, rather than “TG+syn”.
Discussion
Line 379 to 381: In sentence “We successfully demonstrate widespread brain expression of the transgenes via intravenous injection of AAV-PHP.eB in each of our transgenic mouse models, consistent with previous studies 381 [18, 19].”
Could the transgene be mentioned e.g., “…the transgenes α-syn or tau proteins via intravenous….”.
Discussion Paragraph 2 (lines 385-397. Is this really Discussion of results? Could this information along with details about the APP/PS1 mouse (details about this mouse currently lacking) be included in Materials and Methods section.
Line 407: Could transgene affected in Tg-A53T mice be mentioned; i.e., α-syn.
Line 423-4232: Is it the Aβ plaques that potentiate α-syn pathology/aggregation or is it the increased levels of Aβ peptide/oligomers. Plaques
Author Response
We thank Reviewer 3 for the time dedicated to review our study and for the thoughtful comments. See below our point-by-point response to the reviewer's suggestions.
1 - In experimental design, could it be made clearer what each AVV is expressing including AAV-PHP.eB-venus?
R: We have provided clarification to the transgene and venus AAV treatments in the experimental design.
2 - This sentence is confusing - “All mice were tail-vein injected at 3-months of age with 200 μl of the virus and transgene with the same concentrations of viral load (8x1011gc/ml).”
Virus is not a volume (better to say how many viral particles inject in a 200µL volume throughout manuscript) and what’s the difference between virus, transgene and viral load???
Also, what does gc/ml stand for???
R: That is a good point the reviewer raised. We have improved our description of the treatments and dosage, providing clarification to the abbreviation gc/ml (genome copies/milliliter).
3 - Fig 1 C legend: please provide more detail. Age of mice when injected and when post injection were images collected.
R: We have provided more details in the Figure legend.
4 - Why not include 9 MPI data in Figure 2??
R: We did not add the behavioral and immunohistochemical data of 9MPI hTau mice in Figures 2 and 3 because, as shown in the statistical analysis provided in the Supplementary Table 1, the data have not changed significantly from the 6MPI time-point. We believe that adding more graphs and images that show the same effects already described at 6MPI might create confusion and will not fit the current structure of the manuscript. We decided to simply provide the statistical details of the 9MPI point. Cordially, we have provided in the attachment the graphical representation of the statistical analysis of the 9MPI hTau mice to demonstrate that effects at 9MPI remain the practically unchanged from 6MPI.
5 - Could the behavioral test be named at the top of each graph?
We have added the title of each behavioral test to Figures 2, 4, and 6.
6 - Lines 288 and 292: should it be “TG+tau”, rather than “TG+syn”.
Thank you. We have corrected this misidentification.
Discussion
7 - Line 379 to 381: In sentence “We successfully demonstrate widespread brain expression of the transgenes via intravenous injection of AAV-PHP.eB in each of our transgenic mouse models, consistent with previous studies 381 [18, 19].”
Could the transgene be mentioned e.g., “…the transgenes α-syn or tau proteins via intravenous….”.
R: We have improved the description of the transgenes in this section.
8 - Discussion Paragraph 2 (lines 385-397. Is this really Discussion of results? Could this information along with details about the APP/PS1 mouse (details about this mouse currently lacking) be included in Materials and Methods section.
R: We have moved this paragraph to the Animals Section in the Methods and have provided more information regarding the APP/PS1 mice. Thank you for pointing this out.
9 - Line 407: Could transgene affected in Tg-A53T mice be mentioned; i.e., α-syn.
R: We have provided more details about this study and the mouse line and treatment.
10 - Line 423-4232: Is it the Aβ plaques that potentiate α-syn pathology/aggregation or is it the increased levels of Aβ peptide/oligomers. Plaques
R: Thank you, we have made this correction.
